# Liquid Biopsy for Lung Cancer: Up-to-Date and Perspectives for Screening Programs

**DOI:** 10.3390/ijms24032505

**Published:** 2023-01-28

**Authors:** Giovanna Maria Stanfoca Casagrande, Marcela de Oliveira Silva, Rui Manuel Reis, Letícia Ferro Leal

**Affiliations:** 1Molecular Oncology Research Center, Barretos Cancer Hospital, 1331 Rua Antenor Duarte Vilela, Barretos 14784-400, Brazil; 2Life and Health Sciences Research Institute (ICVS), School of Medicine, Campus de Gualtar, University of Minho, 4710-057 Braga, Portugal; 3ICVS/3B’s—PT Government Associate Laboratory, 4710-057 Braga, Portugal; 4Barretos School of Medicine Dr. Paulo Prata—FACISB, Barretos 14785-002, Brazil

**Keywords:** liquid biopsy, lung cancer, biomarkers, precision medicine

## Abstract

Lung cancer is the deadliest cancer worldwide. Tissue biopsy is currently employed for the diagnosis and molecular stratification of lung cancer. Liquid biopsy is a minimally invasive approach to determine biomarkers from body fluids, such as blood, urine, sputum, and saliva. Tumor cells release cfDNA, ctDNA, exosomes, miRNAs, circRNAs, CTCs, and DNA methylated fragments, among others, which can be successfully used as biomarkers for diagnosis, prognosis, and prediction of treatment response. Predictive biomarkers are well-established for managing lung cancer, and liquid biopsy options have emerged in the last few years. Currently, detecting EGFR p.(Tyr790Met) mutation in plasma samples from lung cancer patients has been used for predicting response and monitoring tyrosine kinase inhibitors (TKi)-treated patients with lung cancer. In addition, many efforts continue to bring more sensitive technologies to improve the detection of clinically relevant biomarkers for lung cancer. Moreover, liquid biopsy can dramatically decrease the turnaround time for laboratory reports, accelerating the beginning of treatment and improving the overall survival of lung cancer patients. Herein, we summarized all available and emerging approaches of liquid biopsy—techniques, molecules, and sample type—for lung cancer.

## 1. Introduction

Lung cancer has the highest incidence rate for cancer type and the second-highest mortality rate in the world [1]. The high mortality rates associated with lung cancer are mostly due to its late detection and diagnosis, leading to a decrease in these overall survival rates [2,3]. However, even patients diagnosed in the early stages of the disease and still during cancer’s contained stage still show a poor five-years overall survival rate, which is lower than 60% compared to other cancer types. Despite the early detection, the poor prognosis is attributed to the disease progression [4].

Cigarette smoking and other tobacco exposure habits are the major risk factor for lung cancer, and it is associated with approximately 80% of all lung cancer cases [5,6]. However, additional risk factors associated with environmental and occupational exposures have also been linked to lung cancer development [7]. In addition to environmental risk factors, single nucleotide polymorphisms (SNPs) have also been associated with an increased risk for lung cancer development [8,9,10].

The biological processes behind lung cancer are complex, and the tumors are highly heterogeneous [11]. Histologically, lung cancer is divided into two major subtypes: small-cell lung cancer (SCLC) and non-small-cell lung cancer (NSCLC) [12]. The SCLC accounts for approximately 15–20% of all lung cancer cases, and SCLC patients show extremely low survival rates [13]. The NSCLC accounts for approximately 80–85% of all lung cancer patients, and it is further classified into three main histological subtypes: adenocarcinoma, squamous cell carcinoma (SCC), and large cell carcinoma [14]. NSCLC patients present heterogeneous survival rates depending on several features, including the stage of the disease at the time of diagnosis, the patient’s performance status, smoking habits, histological subtypes, and molecular characteristics [15,16].

In addition to the histological categorization, molecular characteristics are a critical additional component of the biological study of lung cancer due to the advances in precision medicine for both methodological procedural aspects and molecular biomarkers [17]. Currently, the number of FDA-approved drugs targeting molecular biomarkers has dramatically increased in the market, creating a more efficient route for treating NSCLC patients [18]. On the other hand, tumor tissue biopsy has continued to be mandatory for histological evaluation and diagnosis, and in the last decades, it has also been employed in the detection of molecular biomarkers [18,19]. However, the methods commonly used for tissue procurement are highly invasive and may have many side effects, which depending on the performance status of a patient, may not even be eligible for such an invasive procedure. Moreover, tumor tissue biopsy represents a small fragment of the tumor as a whole, which may result in unviable tissue biopsies for assessing tumor heterogeneity [20]. Thus, the development of new approaches is pivotal, given that the current approaches have some disadvantages, especially when it comes to offering a viable option for those who are not eligible for the currently existent methods of sample procurement [20,21,22,23].

In the last few years, a new approach has emerged, attracting various efforts for its implementation in the diagnostic routine of lung cancer [24,25]. Liquid biopsy is a minimally invasive approach for sample procurement, mostly body fluids, and it is used to detect molecular alterations, tumor cells, and metabolites. Furthermore, liquid biopsy, the emerging approach attracting substantial attention, has currently been employed for clinical management, specifically for guiding treatment and monitoring disease. In addition, to being a minimally invasive approach, the liquid biopsy also enables the serial collection of samples allowing early detection of residual disease and relapses and resistance to treatment [20,21,23,26,27]. For NSCLC, this approach is particularly valuable for patients who are not eligible for the conventional tissue biopsy, mainly due to patients’ individual conditions and the tumor location, as previously discussed. As a result, liquid biopsy has been employed for disease and treatment monitoring and for creating a targeted treatment for NSCLC patients within the field of precision medicine [21,28].

This review aims to gather the most up-to-date information for the procedure of liquid biopsy, given the context of NSCLC patients. This review will focus on information such as the various types of body fluids used for liquid biopsy, the currently available and promising biomarkers in the field, and a discussion of the major challenges involved in analyzing and securing samples for liquid biopsy.

## 2. Sample Types and Analytes for Liquid Biopsy from NSCLC

Body fluids such as plasma, sputum, saliva, urine, stool, cerebrospinal fluid, and pleural effusions, among others, are suitable sources for detecting diagnostic, prognostic, and predictive biomarkers for NSCLC [21,29,30,31,32,33]. As a result, liquid biopsy has been employed due to its minimally invasive sample procurement. However, it is important to point out that not all body fluids are collected by a minimally invasive sample procurement (e.g., cerebrospinal fluid). Thus, the type of body fluid selected for liquid biopsy must be carefully chosen based on the type of cancer, especially because some body fluids do not properly represent tumor origin (Table 1 and Figure 1). Moreover, the most important benefit of liquid biopsy is the minimally invasive sample procurement, and we should avail this advantage.

Liquid biopsy samples are mostly composed of cell-free DNA (cfDNA), cell tumor DNA (ctDNA), circulating cell-free microRNAs (ccfmiRNAs), circulating tumor cells (CTCs), metabolites, and proteins as well as extracellular vesicles such as exosomes, which contain proteins and cell-free nucleic acids (cfNA) such as miRNAs (Figure 1). These components are released into body fluids through processes such as apoptosis, necrosis, and secretion [26]. Therefore, liquid biopsy has been an important “supporting actor” for guiding therapeutic strategies for NSCLC patients and has emerged as a “leading title-role actor” in the routine setting of precision medicine.

## 3. NSCLC Biomarkers for Detection in Liquid Biopsy Samples

Clinical biomarkers can be found in different biofluids; currently, the most frequently used in precision medicine are plasma and serum, both of which originate from the peripheral blood [49]. Both blood-derived biofluids can be used to analyze different biomarkers from distinctive sources, such as CTCs, ctDNA, cfDNA, ccfmiRNAs, and metabolites, among others [50]. The major applications of each type of molecule in the field of lung cancer are summarized hereafter.

### 3.1. Circulating-Tumor DNA (ctDNA) and Cell-Free DNA (cfDNA)

Fragments of cell-free DNA (cfDNA) are freely available throughout the blood. CfDNA is released into the bloodstream due to natural body mechanisms, such as apoptosis, necrosis, and active secretion [26,31]. CfDNA can be found in both healthy subjects and cancer patients, although cfDNA levels tend to be higher in cancer patients (Figure 2).

Circulating tumor DNA (CtDNA), different from cfDNA, can harbor somatic mutations reflecting the tumor dynamics [51,52]. Tissue biopsy is a single “snapshot” of the tumor and, therefore, can mirror a unique subclone or a few subclones, while ctDNA can be more representative of the tumor’s entire tissue composition. Moreover, it is currently possible to trace the subclone’s origin of relapses and metastases through the phylogenetic profile of ctDNA. Tracing the clonal origin of the tumor can provide a collage of a tumor’s evolution, opening possibilities for the translation of such information into clinical practice [34].

Using liquid biopsy to detect ctDNA is extremely effective, as it is possible to collect a sample at any time during the disease and therapy course, assessing the progression of the disease in real-time [51]. However, liquid biopsy can become challenging when the concentration of ctDNA is measured to be extremely low (<1%) compared to the concentration of cfDNA in a patient’s bloodstream. Thence, highly sensitive and specific techniques are required for detecting mutations to track, detect, and monitor genomic alterations as cancer progresses [52,53,54]. The current standard methods for detecting somatic mutations, such as RT-PCR, Sanger sequencing, and next-generation sequencing (NGS), may not be sensitive enough for the detection of mutational ctDNA, specifically mutations presented with a low variant allele frequency (VAF) [51].

The main advantage of analyzing ctDNAs is the high specificity of the molecule, as it is proven that any molecular alterations present in these molecules are identical to those in the tumor tissue. In addition, advanced-stage cancer patients usually present with higher levels of ctDNA, which allows for easy monitoring of the disease’s course, tumor heterogeneity and dynamics, tumor evolution, and any tumor-acquired resistance to targeted treatments. Recently, ctDNA was associated with shorter survival, and actionable alterations in ctDNA were not detectable in time-matched tissue [55]. In addition, the turnaround time was decreased for the release of a biopsy report to guide therapeutic decisions that can be made earlier and safer [55].

Currently, highly sensitive PCR-based techniques, such as droplet digital PCR (ddPCR) and BEAMing, are considered sensitive enough for detecting mutations at low frequencies and have emerged as potential tools not only for advanced cancers but also for early detection [53]. NGS is also an extremely sensitive technique for detecting somatic mutations and identifying mutational frequencies as low as 0.02% VAF; however, the NGS technique produces an error rate of about 0.5–2.0% [56].

Therefore, the detection of rare variants by NGS remains challenging due to its limit of detection (LoD) and errors incorporated during sequencing. Furthermore, random errors can be incorporated into the DNA molecules during the library preparation or the sequencing processes, which could be mistaken for true variants. Strategies using molecular barcoding principles, such as hybrid capture and amplicon-based NGS, have also been used to detect low-frequency mutations in both liquid and conventional biopsy samples to decrease false negative results [51,53]. Each molecule in the sequencing library is marked with a small sequence of random nucleotides (8–16 N), which can be called a Unique Molecular Identifier (UMI), Tag Sequencing, or molecular barcodes and to decrease the chances of detecting false variants, the sequenced reads are grouped according to these UMI. As a result, sequencing artifacts can then be detected as they are not present in all reads with the same UMI, which increases the reliability of naming the true variants. Therefore, molecular barcoding technology allows for the correction of sequencing errors [57,58].

Due to the advances in sensitive technologies, ctDNA and mutational analysis are now possible for NSCLC patients. In addition, the detection rate of ctDNA can be higher than 80% in the plasma from NSCLC patients, suggesting that ctDNA analysis is an adequate alternative when sampling tissue biopsy is not an option [59]. *EGFR, KRAS, ERBB2*, and *BRAF* mutations, gene rearrangements (*EML4—ALK*, *ROS1, NTRK1/2*, and *RET*), exon skipping alterations, and gene amplifications (MET) are routinely evaluated in the management of care for NSCLC patients [51]. All these molecular alterations have been adapted into the clinical practice for guiding and monitoring patients’ treatment and disease state [53]. Currently, broader NGS panels have been employed in clinical practices, such as MSK-IMPACT (tissue) and MSK-ACCESS (plasma) [55]. A quarter of the patients presented alterations in ctDNA not detected in tissues [55], which suggests that plasma samples may present a higher specificity then previously reported, supporting the use of NGS as a sensitive, specific tool for plasma samples analysis [60].

Unfortunately, not all of the actionable alterations mentioned above have been translated for the clinical management of NSCLC patients through liquid biopsy sampling.

The FDA recently approved two IVD (in vitro diagnosis) tests (Cobas *EGFR* Mutation Test v2, Roche, and Idylla TM ctEGFR Mutation Assay) for NSCLC patients, which employs plasma samples for the detection of *EGFR* resistance mutation p.(Tyr790Met), exon 19 deletions and p/L858R mutations [61]. Although the IVD test shows a lower sensitivity when compared with other methods of variant detection (e.g., ddPCR and NGS), its approval was a great addition to the present tools used in guiding therapeutic decisions and monitoring treatment results for NSCLC patients [62].

### 3.2. cfDNA Methylation Biomarkers

Genetic mutations and epigenetic modifications, such as DNA methylation, have been detected in cfDNA and ctDNA. It has been considered a promising approach for translation into clinical applications to be used for diagnosis, prognosis, and predictive purposes.

Methylation is an incorporation of a methyl group (CH3) into a Cytosine in regions enriched with CG bases, also known as CpG islands [63]. Methylation occurs in CpG islands when found at the promoter regions of several genes, and it often results in gene silencing commonly found in tumor suppressor genes. On the other hand, the transcriptional activation of genes with the methylation present in the gene body is associated with various cancer types, including lung cancer [41,64]. Methylome analysis has yielded highly successful results on tumor tissues, especially when the analysis is focused on molecular subtyping and biomarkers discovery of several tumor types, including lung cancer [65,66]. In addition to plasma and serum, other body fluids deserve attention, such as sputum and bronchoalveolar lavage fluids, specifically due to their proximity to the tumor’s location [41].

Analysis of methylation-based biomarkers in cfDNA can also be used for diagnostic purposes for the management of lung cancer patients. Plasma cfDNA showed significant differences in DNA methylation level for early-stage NSCLC patients, including stage IA patients, indicating this type of biomarker could be a valuable tool for screening and early detection of NSCLC combined with imaging tests to improve the detection of early-stage pulmonary nodules [63,67,68,69,70]. Multi-cancer early detection (MCED) test may be a promising approach for cancer screening and early detection. MCED is a targeted methylation-based assay for complementary use in screening programs [71,72,73]. However, results for stage I lung cancer have not shown to be sensitive enough for being employed in a screening setting [72]. A recent study screened asymptomatic subjects from the NHS-Galleri trial (ISRCTN91431511) but results about the clinical utility of the MCED test for lung cancer still need to be addressed [73].

The combined methylation analysis of the CDO1 and HOXA9 was associated with unfavorable outcomes, while the combination of PTGDR and AJAP1 methylation was associated with favorable outcomes. A prognostic risk based on these methylation-based biomarkers may be useful to refine risk stratification [65,67]. In addition to prognostication, methylation-based biomarkers can also be useful as predictive biomarkers. Deregulation of methylation in cfDNA was also associated with EGFR-TKI resistance in early-stage NSCLC patients [74,75].

### 3.3. Circulating Tumor Cells (CTCs)

Circulating tumor cells (CTCs) are cells derived from primary tumors that were dissociated from tumor mass by either mechanical motion, the loss of adhesion molecules on the surface of cells entering the circulatory system, or a combination of both. CTCs are commonly detected in lower concentrations in the peripheral blood compared to other types of analytes (e.g., ctDNA), and it may be a challenge requiring the sampling of CTC-rich peripheral blood [38,52,54]. Although studies have shown that aggressive tumors can release thousands of these cells into the bloodstream every day, and it may be associated with the mechanism of distant metastasis, such a result would require completing a complex process [38,54]. Once these cells enter the bloodstream, they are capable of planting metastatic sites through active trans-endothelial migration while remaining inactive; however, no studies have yet been able to show how the biological process of transition from dormancy to active growth happens [52]. As a result, the mechanisms through which cancer spreads from one organ to another using CTCs are of great interest to the scientific community [54].

Furthermore, the methods of isolating CTCs include identifying these cells based on the presence of specific markers, such as the epithelial cell adhesion molecule (Ep-CAM), the cytokeratin of epithelial CTCs, and the N-Cadherin or vimentin of mesenchymal CTCs [76,77]. The FDA has confirmed an IVD method, based on anti-EpCAM ferromagnetic microbeads, called CellSearch CTC kit^®^ (Veridex LLC, Raritan, NJ, USA) for prognostic assessment. This IVD method detects anti-EpCAM ferromagnetic microbeads on circulating tumor cells (CTC) in peripheral blood samples [78]. As previously mentioned, the characterization of CTCs is difficult since the detection method requires high sensitivity to detect these molecules’ low concentrations in extracorporeal fluids.

The presence of CTCs is considered a prognostic biomarker since it can help predict disease progression in cancer patients, including the progression of NSCLC [54]. One study showed that patients with metastatic lung cancer and a state of progressive disease presented a higher expression of the *PIK3CA, AKT2, TWIST*, and *ALDH1* genes in CTCs compared to patients with non-metastatic disease. Thus, it is possible to correlate the presence of CTCs with the diagnostic scope of disease burden, including possible metastasis and disease progression [37]. The presence of CTCs was found to be independent of the tumor stage at diagnosis (49% stage I, 48% stage II, 48% stage III, and 52% stage IV patients) and histology (47% adenocarcinoma and 40% squamous cell carcinoma) when using a size-based filtration method for CTC detection [79]. Moreover, of the cells with the highest glucose uptake, hypermetabolic CTCs were isolated to analyze *EGFR* and *KRAS* mutations using ddPCR. The comparison between the primary tumor’s *EGFR* and *KRAS* mutations and that of CTCs showed a match in 70% of the cases [80].

### 3.4. Extracellular Vesicles

Extracellular vesicles (EVs) can be divided into microvesicles, vesicles, and exosomes. Exosomes are released by various methods and are detectable in several cell types and bodily fluids, including cancer cells. Moreover, exosomes are released during the process of exocytosis following the fusion of multivesicular bodies (MVBs) and cell membranes, and they can also be detected in body fluids such as blood (plasma and serum), urine, pleural effusions, saliva, cerebrospinal fluid, and semen [27,54]. In addition, exosomes have been shown to mediate intercellular communication between the tumor and the stroma, resulting in a rich source of molecular information that enables the location of origin cells for these exosomes [50,81].

Furthermore, ultracentrifugation or differential centrifugation (DC) and size-exclusion chromatography, based on size selection and chemical isolation during polymeric-based precipitation (PBP), are all commonly used for EV isolation [82,83,84].

Tumor cells can release more exosomes than non-tumor cells, making exosomes a potential biomarker in liquid biopsies for various types of tumors [27]. In lung cancer, exosomal RNA, DNA, and proteins can be used to detect molecular alterations, including actionable mutations [27,85]. Qu et al. (2019) identified exosomes harboring EGFR mutation, showcasing exosomes as a valuable component in the analysis’s tumor progression, pre-metastatic niche formation, and resistance to treatment [86]. The comparison between exosome-derived and tumor-derived mutations showed a sensitivity of 100% in detecting EGFR mutations and a specificity greater than 96% [86].

Furthermore, exosomal miRNAs (exo-miRNA) have also been generating interest from the scientific community as a potential biomarker in several types of cancer, including lung cancer. Exo-miRNA is protected from RNAse degradation in the bloodstream due to the lipid bilayer membrane, which results in their stability in body fluids and easily detectable form in body fluids [27]. As a result, exo-miRNA load was reported as a prognostic biomarker [87]. Additionally, exo-miRNAs miR-564 and miR-659 switched sensitive to resistant cells to gefitinib [88], which proved to be a predictive phenotype for exo-miRNA. On the other hand, the exo-miRNA miR-302-b is associated with the suppression of lung cancer cell proliferation and migration via the TGFβRII/ERK pathways, which has established the miR-302-b as a potential therapeutic target for lung cancer patients [89]. Thus, exosomal miRNAs may be employed in the process and measurements of diagnosis, prognosis, and monitoring of lung cancer patients, which reveals the need for further studies of exosomes and their clinical application in the context of precision medicine.

### 3.5. MicroRNA/CircRNA

MicroRNAs (miRNAs) are small RNAs (19–24 nucleotides) responsible for regulating gene expression. However, most miRNAs have unknown biological functions [90,91]. MiRNAs differ from mRNAs due to their stability, small size, and regulatory control in gene expression, rendering them a new generation of biomarkers [40]. Circulating-free miRNAs (cfmiRNAs) have already been reported in body fluids from cancer patients, including plasma, serum, urine, and saliva [92,93,94].

The detection of miRNAs in liquid biopsy samples was reported to distinguish NSCLC patients from healthy subjects paving the way for cfmiRNAs as promising early detection biomarkers [95]. Reis and collaborators (2020) recently reported differential expression of a set of miRNAs in plasma samples from early-stage lung cancer patients [96]. Similarly, plasma samples from early-stage NSCLC patients paired with non-cancer patients were screened for 754 circulating miRNAs, and a highly accurate 24-miRNAs panel was proposed for early detection of lung cancer in addition to the known risk factors [95]. In addition, dysregulation of miRNA expression can also be associated with exposure factors.

For example, the downregulation of let-7i-3p and miR-154-5p was found in serum from smokers, both non-cancer subjects and lung cancer patients. These miRNAs are involved in lung cancer development and progression, rendering these circulating miRNAs a potential role as a diagnostic and prognostic biomarker in lung cancer for tobacco-exposed patients [97,98]. Moreover, high expression of miR-34 and miR34-c in plasma samples from surgically resected NSCLC patients was associated with increased survival [98].

Recently, some publications have explored the potential of circular RNA (circRNA) as a biomarker in liquid biopsy [99]. This is because circRNA are resistant to RNases and other exonucleases due to the lack of final 5’ or 3’ regions [100,101], having a longer half-life than linear RNA [102]. Moreover, its specific tissue expression and stage of development increase this interest as a potential biomarker [98,103,104,105]. Studies show that circRNA expression varies in different tumor types, including lung cancer [104,106,107,108]. In adenocarcinoma patients, circRNA can be over-expressed, such as circ_0013958, which was upregulated [107], and circFARSA, a circRNA derived from exon 5–7 of the FARSA gene, in the plasma of patients with NSCLC compared to controls without cancer [109,110].

In addition, studies with circRNAs reported that they could detect specific signature profiles in tissue samples, capable of distinguishing histological groups in addition to NCLSC patients from healthy ones [105]. However, studies are being carried out to analyze the profile of circRNAs signatures in liquid biopsy samples, such as plasma, exosomes, and lymphocytes, to implement these molecules as diagnostic and predictive biomarkers.

Altogether, the detection of miRNAs/circRNAs in minimally invasive samples has emerged as a promising approach for the early detection of NSCLC to be incorporated in lung cancer screening programs and for prognostic purposes to be incorporated in the clinical management of these patients.

### 3.6. Metabolites and Proteins

In pathological conditions, including cancer, circulating metabolites can be detected in body fluids [45]. For detecting metabolites in biological fluids, Gas chromatography-mass spectrometry (GC-MS), nuclear magnetic resonance (NMR), and high-performance liquid chromatography UV detector (LC-UV) along with software (Metaboanalyst, Metabolite Set Enrichment Analysis [MSEA], Metlin, BioStatFlow and Human metabolome database [HMDB]) have been employed for detecting, processing, and analyzing metabolic data [111].

Lung cancer patients show changes in metabolic pathways compared to non-cancer patients; starch and sucrose metabolism, galactose metabolism, fructose, mannose degradation, purine metabolism, and tryptophan metabolism are among the unbalanced metabolic pathways [43]. In addition, some amino acids such as valine, leucine, and isoleucine are related to stress and energy production, regulating many signaling pathways, such as protein synthesis, lipid synthesis, cell growth, and autophagy, and can be found at higher levels in lung cancer patients compared with non-cancer patients [44]. However, these previously reported data must be clinically tested and reproduced in other series.

Although metabolomics has been a promising approach for managing and monitoring cancer patients, some limitations have still been experienced, such as identifying unknown compounds, detecting cancer-specific metabolites, and standardization of cutoffs [111].

In addition to metabolites, serum proteins have currently been used as cancer biomarkers, such as carcinoembryonic antigen (CEA), cytokeratin 19fragment (CYFRA 21-1), cancer antigen 125 (CA 125), neuron-specific enolase (NSE) and squamous cell carcinoma antigen (SCCA) [112,113]. Immunoassay (e.g., ELISA) and mass spectrometry can detect these serum proteins. In the last few years, considerable efforts have been made to find serum protein biomarkers for the early detection of lung cancer that will potentially be employed in screening programs soon.

Acute-phase reactant proteins (APRPs) are produced in response to inflammation caused by cancer and can also be used as potential biomarkers for the diagnosis of different types of cancer [114]. Lung cancer patients presented higher serum haptoglobin β (HP- β) chain levels than healthy controls. However, patients with other respiratory diseases also presented increased levels of Hp-β chain; thus, medical history, and radiological images should also be considered [115]. Lung cancer patients also presented higher serum amyloid A (SAA), another APRP, compared with healthy controls [116]. With higher levels of SAA1 and SAA2, lung adenocarcinoma patients also presented decreased serum levels of Apo A-1—a protein responsible for removing endogenous cholesterol from inflammatory sites. Thus, SAA1, SAA2, and Apo A-1 could also be considered potential biomarkers for the early detection of lung cancer [117].

In addition, to early detection applications, some proteins involved with metastasis and tumor progression are promising prognostic biomarkers. For example, plasma and pleural effusions from NSCLC patients with high levels of the S100A6 protein—a member of the S100 family with pro-apoptotic function—presented longer survival time compared with S100A6-negative cases [118,119]. On the other hand, NSCLC patients show high serum Cytokeratins (CKs) levels, such as CK 8, 18, and 19, which were associated with unfavorable prognoses [119,120]. In addition, the combined analysis of regulators of actin—calmodulin, thymosin β4, cofilin-1, and thymosin β10—was suitable for predicting patients’ outcomes [121].

The CancerSEEK reported forty-one potential protein biomarkers detectable at least in one out of the eight tumor types. The authors selected the best eight biomarkers for composing the final bead-based immunoassay test (CA-125, CA19-9, CEA, HGF, Myeloperoxidase, OPN, Prolactin, TIMP-1), which were highly effective in distinguishing cancer patients from healthy controls [122]. The positivity for the CancerSEEK test was about 70% for all cancer types. According to this study, combining protein biomarkers with cfDNA mutations increased sensitivity without significantly decreasing specificity [122].

Although proteomic approaches have been promising tools for precision medicine in the lung cancer field, there are some limitations regarding using proteins as biomarkers. Many of these proteins are associated with different tumor types. They present poor sensitivity, are not organ-specific, and can also be detected in non-malignant diseases [112,113,123]. Thus, clinical history and radiological exams should always be considered. Moreover, some proteins are presented with low serum levels, precluding their employment for early lung cancer diagnosis [123,124]. Finally, the analysis of these biomarkers can be influenced by the contamination of intracellular proteins caused by cell lysis during sample procurement and processing. Thus, pre-analytical steps are pivotal [118].

Scalable proteomics has also emerged as a tool for the identification of high-risk subjects for lung cancer. The most recent example is Olink Proteomics, which was created to detect circulating proteins (https://olink.com/, accessed on 16 November 2022). A 36-protein multiplexed assay was developed for risk assessment of lung cancer development, and this set of proteins includes growth factors, tumor necrosis factor receptors, and chemokines and cytokines [125]. Scalable proteomic approaches may not be affordable for low-middle-income countries, but when feasible, they should still be considered as a complementary tool along with LDCT in lung cancer screening programs. However, more studies should be conducted to prove these proteomic approaches are cost-effective for lung cancer screening, especially considering limited resources and minorities.

### 3.7. Autoantibodies against Tumor-Associated Antigens

Autoantibodies against tumor-associated antigens (TAAs) are circulating antibodies to autologous cellular antigens [126,127,128]. Tumor tissues can release cellular proteins, leading to the activation of the immune system and the production of autoantibodies [129,130]. Therefore, cancer patients produce autoantibodies against aberrant or overexpressed proteins produced by these cancer cells [131]. TAAs are stable in the serum and have been studied in several types of tumors, including lung cancer. TAAs can be detected through the immunoenzymatic assay (ELISA) [48,129,132]. Moreover, they can also provide information for early detection and disease monitoring [47,48,132,133,134,135].

In the 90s, Lubin et al. (1995) analyzed the presence of p53 antibodies in the serum of patients with lung cancer [136]. These antibodies were 30% higher in cancer patients than in non-cancer patients, associated with TP53 mutations [136]. TAAs panels have also been developed, combining many different TAAs to improve test sensitivity [48,126,132,137,138]. The panels developed include TAAs for p53, NY-ESO-1, CAGE, GBU4–5, Annexin 1, SOX2, c-Myc, MDM2, NPM1, p16, cyclin B1, among others [48,132,139]. Although the levels of most biomarkers increase proportionally to tumor burden, TAAs levels do not differ among different stages of lung cancer, possibly due to the humoral immune response—present from the beginning of tumorigenesis [48,132,139]. This feature renders TAAs a potential tool for early detection [48,132,140,141]. However, studies including proper sample size and validation set are required for greater reliability of TAAs in the routine lung cancer setting.

## 4. Clinical Application of Liquid Biopsy for NSCLC

The absence of pathognomonic symptoms in lung cancer leads to late diagnosis—many patients may mistakenly receive another diagnosis, such as pneumonia, Chronic obstructive pulmonary disease (COPD), among others, especially from low-middle income countries, where resources are limited, and health systems are recurrently offscourings. The late diagnosis in limited therapeutic options and curative intent is no longer available, culminating in the highest lethality rate among all cancer types [3].

### 4.1. Prognosis

Lung cancer in the early stage has an excellent prognostic after surgery due to early screening but needs more information about the disease is needed to understand metastatic lung cancer [142]. After the liquid biopsy advent, the prediction of outcomes for NSCLC patients employing molecular biomarkers became more feasible for those who may not be eligible for conventional biopsy [21,27].

In NSCLC and SCLC, more CTCs were associated with adverse prognostic factors for survival [54,143,144]. A study compared the number of CTCs present in blood in patients with small-cell lung cancer. The CTC count was made at the beginning, before, and after treatment. The authors show that patients with eight or more CTCs, per 7.5 mL of blood had a worse survival than those with less than 8 CTCs at the pretreatment, and patients whose baseline CTC levels remained lower than 8 CTC at the posttreatment showed better survival [143].

Undetectable ctDNA blood levels may serve as a prognostic marker for targeted therapy and chemotherapy in patients with NSCLC [142,145]. CtDNA has been proposed as a non-invasive, real-time biomarker to provide prognostic information to monitor treatment since patients with the same mutation may differ in response to treatment [146]. Based on this, techniques can detect recurrent point mutations in controller genes. When using ddPCR, plasma G12/G13 status was associated with an unfavorable prognostic in progression-free survival and overall survival in NSCLC patients [147]. The prognostic value of TP53 in lung cancer is being debated. Several changes in this gene are found in patients with advanced NSCLC. Patients carrying pathogenic plasma mutations in TP53 have lower overall survival when compared to wild TP53. In addition, the risk of extrathoracic metastases in patients with TP53 ctDNA alteration is greater [148].

In addition to point mutations, the methylation status of critical genes can also be analyzed in minimally invasive samples, and the methylation status can serve as a prognostic biomarker [135,142]. For example, Wen et al. (2021) showed that circulating metHOXA9 is an adverse prognostic factor in patients with advanced NSCLC. In this study, the authors analyzed the levels of metHOXA9 in patients’ blood before starting treatment and before each cycle of chemotherapy. As a result, it was observed that the levels of metHOXA9 increased after the first treatment cycle; that is, the treatment changed the status of the biomarker, decreasing the overall survival of these patients [149].

The hypermethylation of some genes as p16, CDH1, FHIT, and APC, can be considered a prognostic factor. In addition to being classified by stages and histological type, genetic alterations among each stratified group may reveal additional and more accurate prognostic factors [150]. Such as, the methylation of genes p16, CDH13, RASSF1A, and APC in patients with early-stage NSCLC-treated surgery was associated with an early recurrence [151,152]. More studies are needed to analyze the best genes to use as prognostic biomarkers in lung cancer. By considering post-transcriptional mechanisms, miRNAs can also serve as prognostic biomarkers. For example, NSCLC patients with downregulated miR-590-5p had significantly lower median survival rates when compared to patients expressing high miR-590-5p, and it was associated as a potential prognostic marker for the progression of NSCLC [152].

High expression levels of miR-18a, miR-20a, miR- 92a, miR-126, miR-210, and miR-19a correlated with worse disease-free survival (DFS), in addition to a shorter overall survival (OS) than patients with low expression levels of these miRNAs. So, these results suggested that these miRNAs can be potential biomarkers for the prognosis of NSCLC patients [153]. In addition, elevated levels of miR-34a correlated with a prolonged DFS and OS compared to low levels of expression in 196 NSCLC patients, i.e., miR-34a has potential prognostic value for lung cancer patients [98].

Boeri et al. (2011) and Tian et al. (2016), with a short cohort, reported that miR-486-5p was downregulated in adenocarcinoma patients’ plasma, and miR-181b-5p was upregulated in squamous cell carcinoma patients’ plasma. In addition, the two miRNAs negatively regulate their targets—RASSF1 and PIK3R1 [148,149], while that elevated levels of miR-21 in plasma samples predicted poorer overall survival of NSCLC patients [154,155]. However, more studies with increased sample sizes must incorporate microRNAs into clinical practice as prognostic biomarkers [154].

### 4.2. Precision Medicine and Disease Monitoring

Positron Emission Tomography/Computed Tomography (PET/CT) and tissue biopsy are currently employed for NSCLC disease staging and monitoring and for guiding therapeutic strategies either based on disease staging or molecular alterations [43,156]. However, not all patients are eligible for tissue biopsy, depending on tumor location and the patient’s clinical performance [157]. Using liquid biopsy for clinical management, including treatment and disease monitoring, can better represent tumor heterogeneity, and predictive biomarkers can be successfully detected to guide therapeutic options for NSCLC. Unfortunately, the successful detection of predictive biomarkers in this sample depends on assay sensitivity and the variant allele frequency (VAF). Fortunately, susceptible technologies have emerged as suitable approaches for cfDNA analysis, making rare detecting alleles feasible [126,158,159].

There are two approved tests for the search for actionable mutations in liquid biopsy, the Idylla TM ctEGFR Mutation Assay and the Cobas® EGFR Mutation Test v2, equally real-time PCR-based [160,161]. Identifying EGFR mutations has become essential since EGFR-TKI therapy has become the standard treatment choice for EGFR mutant patients [162]. The acquisition of resistance to treatment is recurrent in up to 60% of patients, and disease recurrence undergoing EGFR-TKI therapy is generally mediated by the p.(Tyr790Met) mutation [163]. As Cabanero and Tsao (2018) suggest, in patients diagnosed with NSCLC, ctDNA analysis can provide tumor resistance responses acquired in real-time to TKIs for EGFR [164]. For them, the clinical application of these tests is about to reach reality because the current digital platforms approach greater sensitivity and precision to ctDNA [165]. Some patients presented with the EGFR p.(Tyr790Met) mutation early as 344 days before disease recurrence [160]. EGFR-mutated patients presented undetectable ctDNA after four weeks of TKi therapy, associated with a 12-week radiological response and progression-free and overall survival [66].

A prospective study was carried out in 2015 using available serial samples of blood—collected and follow-up for ten months—of 41 patients with lung cancer. All samples of blood were analyzed cfDNA observing EGFR mutation and p.(Tyr790Met) mutation, and the authors observed that the appearance or increase in a unit of the p.(Tyr790Met) allele frequency almost triples the risk of death and progression, and this information can be used for to estimate whether p.(Tyr790Met) positive patients should start second-line treatment based on molecular data rather than imaging data [166].

In addition to mutations, the decreased concentration of CTCs has been associated with the radiographic response of the tumor during the different treatments in patients with NSCLC (surgery, chemotherapy, radiotherapy, target therapy, immunotherapy), while the increased number of CTCs has been correlated with the disease progression [167].

Studies have also investigated the expression of PD-L1 in CTC and white blood cells (WBC) in NSCLC [68,168,169,170]. Expression of PD-L1 in CTC and WBC was highly correlated with the tumor tissue expression, pointing out the importance of this evaluation of the “liquid microenvironment” to assist in the immunotherapy stratification and the monitoring of disease [170]. Studies with tumor tissue from patients with NSCLC have shown a relationship between high tumor mutational burden (TMB) and a longer duration of response and survival in patients treated with anti-PD-1 or anti-PD-L1 therapy [171,172,173]. For example, Gandara et al. (2018) analyzed 797 plasma samples from NSCLC patients, and the authors observed that TMB in plasma (bTMB)≥ 16 had a higher benefit for progression-free survival with atezolizumab therapy [174]. However, few studies have been conducted with a blood-based assay to measure bTMB in lung cancer, and the effectiveness of bTMB remains unclear [175].

Gene fusions can also be detected in blood samples of NSCLC patients, such as anaplastic lymphoma kinase (ALK, ROS1, RET, and NTRK) [176]. The detection and permanence of this fusion were associated in this study with shorter progression-free survival to crizotinib [177].

### 4.3. Early Detection as an Emerging Application of Liquid Biopsy for NSCLC

The high mortality rate of lung cancer patients is related chiefly to late diagnosis when curative treatments are ineffective. In order to decrease lung cancer mortality, screening programs have been implemented worldwide [178,179].

There are currently cancer screening programs, such as breast cancer and lung cancer, which use X-ray and low-dose computed tomography (LDCT) emission, respectively, to increase the chances of early diagnosis and increase patient survival. However, these tests are performed only in the percentage of the population considered at high risk; that is, the young do not have the opportunity to undergo the screening procedure because they also have a chance of developing cancer. In addition, developing countries, associated with low income and the lack of easy access to public health programs, make screening programs difficult to occur due to the cost of equipment and the need for mobile units, which lead to late diagnosis of cancer [180]. For this reason, implementing public policies to aid research and development in the context of liquid biopsy is significant.

However, many challenges have been experienced in lung cancer screening programs, and additional strategies are required to increase the early detection of lung cancer [181]. Plasma levels of cfDNA from NSCLC patients are higher than controls with no cancer, rendering cfDNA plasma levels an exciting approach as a diagnostic biomarker [107,182]. CTCs can be eliminated by the primary tumor, even in the early stages of tumor development. However, they are available at low abundance, requiring precision and empathetic methodologies to detect them [183]. Improvements in the CTC isolation methods have emerged, making CTCs in clinical practice feasible to increase the patient’s possibility of care and quality of life [76]. Moreover, the use of biomarkers to distinguish benign from malignant lesions, and to identify molecules that can complement imaging tests during lung cancer screening, can reduce the number of false positives and false negatives [179,180].

MiRNAs released in plasma and serum may also be used for lung cancer screening [184]. The use of microRNAs for lung cancer screening and early detection should be considered to be used along with CT images in order to improve the accuracy of screening programs when CT images do not show clinically detectable disease. The Multicenter Italian Lung Detection (BioMILD) trial showed a miRNA signature classifier (MSC) combined with CT was more effective for risk stratification than only CT or only MSC [185]. Subjects with CT positive and MSC negative (CT+/MSC-) showed an HR of 13.73, and subjects with CT positive and MSC positive (CT+/MSC+) showed an HR of 30.71. However, validation and reproducibility studies are necessary for the implementation of this strategy proposed by the BioMILD trial. In addition, the employment of miRNAs as a minimally invasive tool together with CT in lung cancer screening programs still needs to be better addressed, especially due to conflict data about these biomarkers in blood samples [186].

Technologies are under development to improve the detection of rare and/or small molecules present in the human body. circRNAs-based signatures have emerged as a promising tool for the early detection of lung cancer [187,188].

The CancerSEEK improved the early detection of several types of tumors, including lung cancer, by combining several biomarkers in plasma [123]. The authors analyzed plasma samples from 1005 patients with eight different tumor types, and none of the patients received neoadjuvant chemotherapy nor harbored evident metastases before blood collection. CtDNA isolated from plasma for mutation analysis was initially submitted to PCR amplification using primers designed to amplify regions of interest in the 16 genes. Simultaneously, eight proteins were analyzed in the same samples by the Luminex bead-based immunoassays technique (Millipore, Bilerica, NY). The accuracy of the prediction of this approach was 39% for lung cancer. The sensitivity of CancerSEEK for lung cancer was approximately 60% [123].

The INTEGRAL Risk Biomarker and Nodule Malignancy project analyzes a proteomic panel, based on Olink assay, with 21 protein relevant for lung cancer using a minimum quantity of blood (<50 uL) to optimize LDCT screening [189]. This project was initiated with case-control cohorts—training and validation sets, totalizing more than four thousand people—of several screening programs, main in EUA, Australia, Singapura, and Canada. However, we expect results from this trial to prove the effectiveness of using proteomics panels in lung cancer screening scenarios.

On the other hand, a multiple cancer early detection test (MCED) developed a biomarker based on cfDNA methylation in the plasma of 4077 subjects. This biomarker showed an overall sensitivity and specificity of 51.5% and 99.5%, respectively [72]. For lung cancer, this same methylation panel has shown a sensitivity of 79.5% in cases with stage II (95% CI).

The DETECT-A study (Detecting cancers Earlier Through Elective mutation-based blood Collection and Testing) was performed with 10,000 women, 65 to 75 years old, with no personal history of cancer, and was analyze mutations of 16 genes in cfDNA and nine protein highly validated biomarkers in blood samples [190]. Twenty-six cancer patients were detected by blood testing, and nine were lung cancer. In addition, a PET-CT scan was performed for 15 patients to exclude distant metastasis [190]. Only 1.2% of the individuals tested in the blood test were submitted to PET-CT, decreasing the costs of imaging scans [190]. These data demonstrate the possible relevance of combining multi-cancer blood testing with PET-CT in the clinical routine, improving early detection of lung cancer in a non-invasive way [191]. In addition, lung cancer screening programs can shift the diagnosis scenario from metastatic to early-stage disease. Although overdiagnosis remains a concern, a recent study reported no increase in the overall incidence rate of lung cancer but a decrease in metastatic cases and an increased number of stage I cases compared with an unscreened population [192]. Furthermore, the development of alternative techniques for screening and early detection, such as liquid biopsy, can help the conventional methods already used, reducing the costs of public coffers and reducing exposure to radiation and patient discomfort.

## 5. Challenges and Limitations

Liquid biopsy has been a promising approach for detecting biomarkers in NSCLC patients. This minimally invasive approach better represents tumor heterogeneity and can also be effective for lung cancer screening.

The lack of standardization prevents liquid biopsy implementation in the clinical routine. For this reason, more studies involving protocol standardization and a more significant number of cases become necessary to obtain a more excellent population representation, generating accurate and applicable results. Another limitation is that some biomarkers are fragile, requiring great pre-analytical care. The interaction between genetics and environment is challenging to control. Additionally, specific and sensitive methodologies are necessary to isolate and analyze these biomarkers, mainly due to the low concentration of some molecules in the body fluid.

## 6. Open Issues e Future Perspectives

Although many studies and advances have been made over the last few years concerning liquid biopsy, little has been translated into clinical practice, probably due to the several challenges already mentioned in this review. In addition, we must consider that the vast majority of studies analyze populations of European and North American, opening a gap for the genomic analyses referring to more heterogeneous populations, such as the population of South America and Africa [193,194].

The development and improvement of specific techniques, both to isolate analytes from liquid biopsy and to analyze them, are necessary to increase the sensitivity and specificities of the tests so that they are safe to be used for both early detection, prognosis, and monitoring. Methodologies that increase confidence to detect rare variants or allow a small initial sample input is welcome in this area. Training professionals in this area is also critical for reliable results. In addition, improving new mathematical and computational methods based on machine learning can also improve liquid biopsy methods making this approach even closer to the routine setting.

In this review, we summarized promising biomarkers that can be used for lung cancer screening associated with gold standard methods—such as LDCT—improving detection rates of screening programs (Figure 3). In the near future, a liquid biopsy will hopefully increase the early detection of lung cancer.

## Figures and Tables

**Figure 1 ijms-24-02505-f001:**
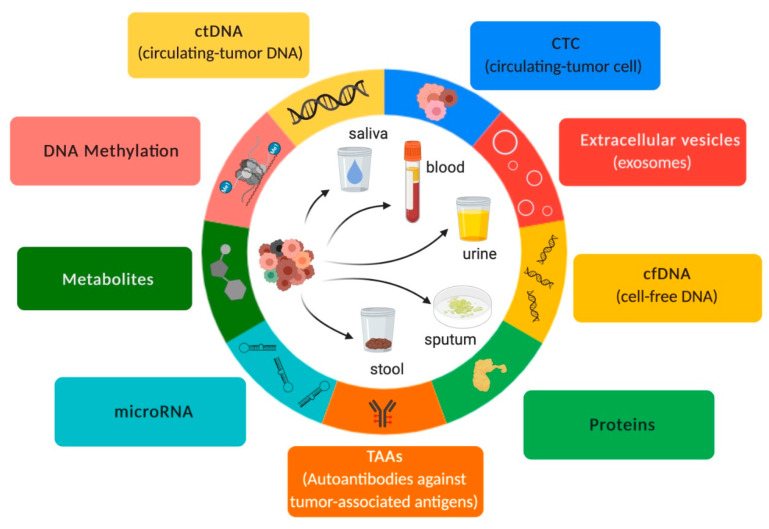
Overview of liquid biopsy. Inspired by Hanahan & Weinberg (192).

**Figure 2 ijms-24-02505-f002:**
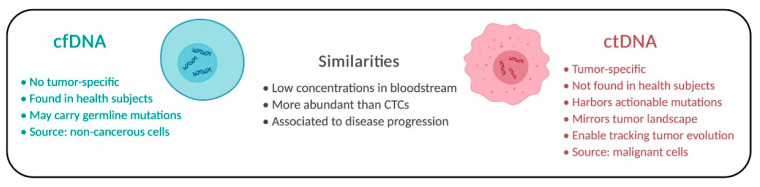
Comparison of cfDNA and ctDNA. Individual aspects and similarities of both molecules are described.

**Figure 3 ijms-24-02505-f003:**
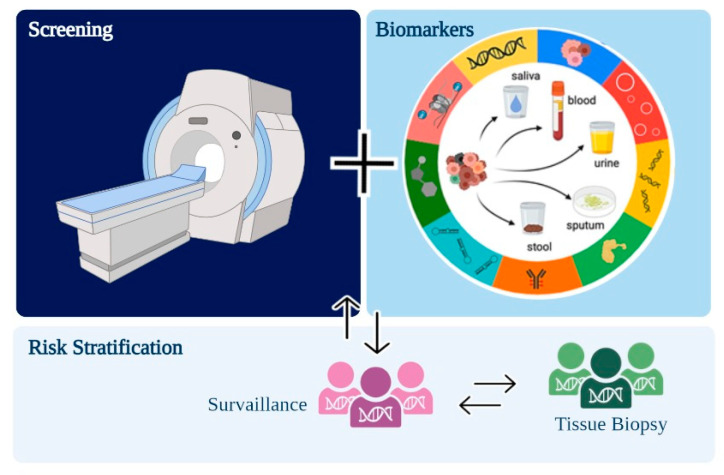
Purposed workflow for lung cancer screening programs associating low-dose computed tomography (LDCT) and minimally invasive biomarkers.

**Table 1 ijms-24-02505-t001:** Liquid biopsy analytes for NSCLC: major clinical applications, biofluids, and methods.

Clinical Application		Biofluids	Methodologies	Reference
ctDNA/cfDNA	Diagnosis, Tumor burden, Treatment response, Prognosis.	Peripheral blood, Sputum.	qPCR, dPCR, ddPCR, ARMS, BEAMing	[32,33,34,35]
CTCs	Diagnosis, Tumor burden, Prognosis.	Peripheral blood	RT-qPCR, Ep-CAM, NGS	[36,37,38]
Extracellular vesicles (exosomes)	Diagnosis, Prognosis.	Peripheral blood	ultracentrifugation, exosomes immunoprecipitation, immune beads precipitation	[39]
miRNAs	Diagnosis, Disease, Progression.	Plasma, Serum, Sputum.	RT-qPCR	[40]
DNA Methylation biomarkers	Diagnosis, Disease progression.	Plasma	Immunoprecipitation, methyl-sensitive restriction enzymes, sodium bisulfite conversion, q-PCR, and Next-Generation Techniques	[41,42]
Metabolites/Proteins	Progression, Predictive, Diagnosis, Prognosis.	Serum	HRMAS MRS UPLC–MS and immunoradiometric assay	[43,44,45,46]
Autoantibodies tumor-associated antigens	Diagnosis, Predictive.	Serum	ELISA	[47,48]

ARMS: Scorpion amplification-refractory mutation system; qPCR: quantitative Polymerase Chain Reaction; dPCR: digital Polymerase Chain Reaction; ddPCR: droplet digital Polymerase Chain Reaction; BEAMing: beads, emulsion, amplification, and magnetics; Ep-CAM: Epithelial cell adhesion molecule, NGS: Next-Generation Sequencing; HRMAS: High-Resolution Magic Angle Spinning; MRS: magnetic resonance spectroscopy; UPLC–MS: ultra-performance liquid chromatography-tandem mass spectrometry.

## Data Availability

No original data were generated by the current manuscript. All information were gathered from open access and/or publicly available manuscripts.

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
