# Peer review of "Liquid Biopsy for Lung Cancer: Up-to-Date and Perspectives for Screening Programs"

_ijms, 2023, doi:10.3390/ijms24032505_

Round 1

Reviewer 1 Report

The work is well organized and in depth regarding all applications of liquid biopsy in NSCLC cancers. It requires a huge revision of the entire bibliography as the references are often inappropriate and do not correspond to the work described. Many are absent in the text and the order is not consecutive as required. References of table 1 are not present in the text. English minor spell check is required

Author Response

Thank you for the response to our manuscript currently entitled Liquid Biopsy in Lung Cancer: Up-to-date and Perspective for Screening Programs”. We also thank reviewers’ insightful comments and pertinent questions. The manuscript has been reviewed and revised with the reviewer’ suggestions incorporated. All modifications in the manuscript are highlighted in yellow (Please see the attachment). Please find below a detailed point-by-point response to the reviewers’ comments. We hope to have satisfactorily addressed the comments.

Comment: The work is well organized and in depth regarding all applications of liquid biopsy in NSCLC cancers. It requires a huge revision of the entire bibliography as the references are often inappropriate and do not correspond to the work described. Many are absent in the text and the order is not consecutive as required.

Reply: We firstly thank you for your kind review. We sincerely apologized for references problem, there was a bug in the reference manager software. All references were revised and adjusted throughout the text.

Comment: References of table 1 are not present in the text. English minor spell check is required

Reply: Thak you for your observation. References in table 1 were misleading and they were all adjusted accordingly.

Reviewer 2 Report

Thank you for the chance you gave me to read this interesting study entitled “Liquid Biopsy for Lung Cancer: State-of-the-Art” by Casagrande et al. In this review paper, the authors give an overview of the current understanding as well as the clinical significance of liquid biopsy in lung cancer. Although, this topic has great importance and the manuscript is well-written, however, there are many significant issues with this manuscript, including novelty, which need to be treated before being suitable for publication.  I think that this study in the current form does not satisfy the appropriate criteria for publication.

Major points:

Regarding novelty, the same topic has been reviewed by numerous other groups during the last year (e.g. https://doi.org/10.1038/s41416-022-01777-8, https://doi.org/10.1186/s12943-022-01505-z, https://doi.org/10.3390/clinpract12030046, https://doi.org/10.3389/fonc.2021.801269). According to my point of view, the authors should interpret the published studies in a more creative way focusing on less studied points of this issue.

The title of the paper should be changed since it is almost similar to that of previous review studies (e.g. https://doi.org/10.3390/cancers13163923,

According to the score (46%) obtained by the plagiarism detection service “Turnitin”, the manuscript needs to be modified in some parts in order this score to be reduced.

It would be very useful for the readers the authors to provide tables with the studies which have been included in the current review paper focusing on specific issues (e.g. ctDNA)

Author Response

Thank you for the response to our manuscript currently entitled Liquid Biopsy in Lung Cancer: Up-to-date and Perspective for Screening Programs”. We also thank reviewers’ insightful comments and pertinent questions. The manuscript has been reviewed and revised with the reviewer’ suggestions incorporated. All modifications in the manuscript are highlighted in yellow (Please see the attachment). Please find below a detailed point-by-point response to the reviewers’ comments. We hope to have satisfactorily addressed the comments.

Comment: Regarding novelty, the same topic has been reviewed by numerous other groups during the last year (e.g.https://doi.org/10.1038/s41416-022-01777-8, https://doi.org/10.1186/s129430221505z,https://doi.org/10.3390/clinpract12030046,https://doi.org/10.3389/fonc.2021.80126). According to my point of view, the authors should interpret the published studies in a more creative way focusing on less studied points of this issue.

Reply: Dear referee, we sincerely appreciate your careful revision. We acknowledge your point of view, and we agree that writing the manuscript in a more creative way certainly improved this review paper. Different from the above-mentioned review articles, we focused on liquid biopsy for screening and early-detection. We hope to have addressed your suggestion. By the way, a new figure was designed (Figure 3) to provide a take-home message from our review manuscript.

Comment: The title of the paper should be changed since it is almost similar to that of previous review studies (e.g. https://doi.org/10.3390/cancers13163923)

Reply: Thak you for your suggestion. We change the title for “Liquid Biopsy of Lung Cancer: Up-to-date and Perspective for Screening Programs”.

Comment: According to the score (46%) obtained by the plagiarism detection service “Turnitin”, the manuscript needs to be modified in some parts in order this score to be reduced.

Reply: We a acknowledge your concern. This manuscript was written and revised by all authors and any similarity was probably at randomly. Even though, we applied a software for plagiarism check called “CopySpider”. Based on the alghoritm from CopySpider, we suppose it can provide a more fairer review for plagiarism. According to CopySpider, plagiarism rate of the current version of the paper was 1.22%, which is below limit rate (3%).

Comment: It would be very useful for the readers the authors to provide tables with the studies which have been included in the current review paper focusing on specific issues (e.g. ctDNA)

Reply: Thank you for your suggestion. Table 1 was designed for with purpose

Round 2

Reviewer 1 Report

Bibliography has been extensively revised but I think the 149 ref in lane 476 is not correct (check 148 and 149 reffs)

The term "thushus" in lane 383 should be modified